# MicroRNAs as a Suitable Biomarker to Detect the Effects of Long-Term Exposures to Nanomaterials. Studies on TiO_2_NP and MWCNT

**DOI:** 10.3390/nano11123458

**Published:** 2021-12-20

**Authors:** Sandra Ballesteros, Gerard Vales, Antonia Velázquez, Susana Pastor, Mohamed Alaraby, Ricard Marcos, Alba Hernández

**Affiliations:** 1Group of Mutagenesis, Department of Genetics and Microbiology, Faculty of Biosciences, Universitat Autònoma de Barcelona, 08193 Cerdanyola del Vallès, Spain; sandra.ballesteros@uab.es (S.B.); antonia.velazquez@uab.es (A.V.); susana.pastor@uab.es (S.P.); mohamed.alaraby@science.sohag.edu.eg (M.A.); 2Finnish Institute of Occupational Health, 00250 Helsinki, Finland; gerard.vales@ttl.fi; 3Zoology Department, Faculty of Sciences, Sohag University, Sohag 82524, Egypt

**Keywords:** microRNAs, oncogenesis, carcinogenesis, cell transformation, nanomaterials, long-term exposures, TiO_2_NP, MWCNT

## Abstract

The presence of nanomaterials (NMs) in the environment may represent a serious risk to human health, especially in a scenario of chronic exposure. To evaluate the potential relationship between NM-induced epigenetic alterations and carcinogenesis, the present study analyzed a panel of 33 miRNAs related to the cell transformation process in BEAS-2B cells transformed by TiO_2_NP and long-term MWCNT exposure. Our battery revealed a large impact on miRNA expression profiling in cells exposed to both NMs. From this analysis, a small set of five miRNAs (miR-23a, miR-25, miR-96, miR-210, and miR-502) were identified as informative biomarkers of the transforming effects induced by NM exposures. The usefulness of this reduced miRNA battery was further validated in other previously generated transformed cell systems by long-term exposure to other NMs (CoNP, ZnONP, MSiNP, and CeO_2_NP). Interestingly, the five selected miRNAs were consistently overexpressed in all cell lines and NMs tested. These results confirm the suitability of the proposed set of mRNAs to identify the potential transforming ability of NMs. Particular attention should be paid to the epigenome and especially to miRNAs for hazard assessment of NMs, as wells as for the study of the underlying mechanisms of action.

## 1. Introduction

The nanotechnology industry is constantly producing new nanomaterials (NMs) with potential applicability in many fields. Consequently, their presence in our environment has substantially increased in recent years [1]. Due to their high surface-to-volume ratio, NMs show high biological reactivity when interacting with different cellular molecules and therefore have the potential to induce adverse effects on any exposed organism, including humans [2]. Accordingly, it is necessary to investigate the potential toxic, genotoxic, and cumulative effects of NMs, as well as to develop new biomarkers for the correct assessment of their possible negative effects on both humans and the environment [3].

Although many experimental models and biological endpoints have been used to evaluate the harmful effects of NMs, few studies have evaluated their potential carcinogenic risks using long-term in vitro approaches. Since carcinogenesis is a complex, multi-step process, prolonged exposures over time are necessary to induce cell transformation. Hence, a long-term exposure approach is required when evaluating the carcinogenic potential of a given compound. However, such an experimental exposure scenario is not frequently found in NM testing studies since most studies focus on short-term in vitro exposures using high doses of the compound.

It has already been demonstrated that long-term in vitro exposures to non-cytotoxic doses of NMs are able to induce the acquisition of different hallmarks of cancer in exposed cells, such as morphological cell changes, secretion of matrix metalloproteinases, anchorage-independent cell-growth capacity, and migration ability, as observed after exposures to cobalt nanoparticles [4], zinc oxide nanoparticles [5], silver nanoparticles [6], and multi-walled carbon nanotubes (MWCNT) [7]. On the other hand, emerging evidence indicates that NM exposure can cause different epigenetic changes correlated with gene-expression alterations [8,9,10,11]. Among them, changes in the level of microRNA (miRNA) expression have been associated with different pathological processes [12,13]. Interestingly, different studies have demonstrated that many cancers have alternative miRNA expression profiles when they are compared to normal tissues. This indicates that initiation, proliferation, and control of the apoptotic program of tumor cells are modulated by different epigenetic entities, including miRNAs [14]. In this direction, we have recently shown that following long-term exposure to nanoceria, bronchial epithelial BEAS-2B cells acquire an oncogenic phenotype characterized by an increased cell-invasion capacity and tumorsphere-formation ability [15]. Interestingly, we demonstrated that these oncogenic changes were accompanied by altered expression of several miRNAs with known roles in the carcinogenesis process [15]. More specifically, expression levels of about the 30% of an initial panel of 33 candidates showed relevant changes, which included miRNAs considered oncogenes (*OncomiRs*) or tumor suppressors (*Anti-OncomiRs*), based on whether they target tumor suppressor genes or oncogenes, respectively [16]. From that work, we proposed that the panel of miRNAs could be used as biomarker for the evaluation of the carcinogenic potential of NMs.

Here, to reinforce our proposal and further explore the connection between the acquisition of the oncogenic phenotype and the altered miRNA profile, we evaluated the expression levels of the 33 miRNAs from our panel in BEAS-2B cells following long-term exposure to titanium dioxide nanoparticles (TiO_2_NP) or MWCNT for 6 weeks. After exposure to both NMs, BEAS-2B cells exhibited an oncogenic phenotype according to the results obtained in a soft-agar assay. This assay measures anchorage-independent cell-growth capacity, one of the main characteristics of transformed cells [17,18]. BEAS-2B cells were selected as the target since inhalation is the most likely route of unintentional human exposure to different NMs, including TiO_2_NP and MWCNT. Both NMs have been extensively used, and their environmental consequences and potential effects on humans have been recently reviewed [19,20].

This study moves one step further than the previous one [15] by identifying a reduced set of informative miRNAs useful in detection of potential oncogenic effects induced by NM exposures. Since miRNAs participate in many overlapping cellular processes, the miRNA-level changes of the proposed set can also be conceived as more general biomarkers for toxicological assessment of NMs under scenarios of long-term exposure. Interestingly, this set of miRNAs was successfully validated in other long-term exposed cell systems previously transformed by cobalt nanoparticles (CoNP), zinc oxide nanoparticles (ZnONP), mesoporous silica nanoparticles (MSiNP), and cerium dioxide nanoparticles (CeO_2_NP).

## 2. Materials and Methods

### 2.1. Nanomaterials Characterization

TiO_2_NP (NM-102) and MWCNT (NM-401) were obtained from the repository of the EU Joint Research Centre (Ispra, Italy) in the frame of the EU NanoReg project. Further characterization of these materials was performed using transmission electron microscopy (TEM, JEOL JEM-2011 instrument) (Jeol LTD, Tokyo, Japan).) to determine dry size and morphology. The mean sizes were determined by measuring 100 randomly selected nanoparticles in several TEM images using the ImageJ program. Specifically, the mean size of MWCNT was calculated by measuring the width of isolated fibers. Dynamic light scattering (DLS), and laser Doppler velocimetry (LDV) methodologies (Malvern Zetasizer Nano-ZS zen3600 instrument) (Malvern, UK). were used to determine hydrodynamic size and zeta potential, respectively. For NMs dispersion, TiO_2_NP and MWCNT were pre-wetted in 0.5% ethanol and dispersed in 0.05% bovine serum albumin (BSA) in Milli-Q water. Afterward, the NMs were sonicated for 16 min to obtain a stock dispersion of 2.56 mg/mL, according to the NanoGenotox protocol [21]. For TEM and Zetasizer measurements, the stock suspension was dispersed in water and in culture medium, respectively.

### 2.2. Cell Culture and Exposure Conditions

BEAS-2B cells were cultured as a monolayer in T75 flasks with DMEM medium (Gibco, Paisley, UK) supplemented with 10% fetal bovine serum (FBS; Pasching, Austria), 1% non-essential amino acids (Pasching, Austria), and 2.5 μg/mL Plasmocin (InvivoGen, San Diego, CA, USA) and incubated at 37 °C in a humidified atmosphere of 5% CO_2_. During the 6-week duration of the long-term exposure, sub-confluent cells were passaged weekly at a cell density of 5 × 10^5^ per T75 flask, and the medium was changed every two days by removing the old medium, then washing the cells twice with phosphate-buffered saline (PBS), then adding new fresh medium containing the NM exposure. Exposures consisted of non-cytotoxic concentrations of 10 and 20 μg/mL of TiO_2_NP and MWCNT, equivalent to 1.34 and 2.67 µg/cm^2^. Importantly, non-treated, time-matched controls were maintained in parallel during the complete period of exposure. All conditions were performed in triplicates for each of the three experiments carried out.

To determine the toxicity of the long-term exposures, a viability assay was performed using the Beckman counter method (Beckman Coulter, Brea, CA, USA). The average number of viable cells for each exposure time point was compared to time-matched, non-exposed controls to calculate cell viability percentages.

### 2.3. RNA Extraction and Retrotranscription Experiments

Changes in the miRNA level of expression were evaluated by real-time RT-PCR. To proceed, total RNA was isolated from long-term exposed and time-matched controls = BEAS-2B cells using TRI Reagent^®^ (Invitrogen, Waltham, MA, USA), in triplicate. RNA quantity was measured on a Nanodrop spectrophotometer (Thermo Fisher Scientific Technologies, Walthan, MA, USA). RNase-free DNase I (Turbo DNA free™ kit, TermoFisher Scientific) was used to remove DNA contamination. An amount of 80 ng of total RNA in a final volume of 10 µL was used for cDNA synthesis. These 10 µL included 1 µL of 10X poly(A) polymerase buffer, 10 mM of ATP, 1 µM of RT-primer (Sigma-Aldrich, Steinheim, Germany), 0.1 mM of each deoxynucleotide (dATP, dCTP, dGTP, and dTTP) (VWR International, Ballicoolin, Dublin, Ireland), 100 units of MulV reverse transcriptase (New England Biolabs, Ipswich, MA, USA), and 1 unit of poly(A) polymerase (New England, Biolabs, Ipswich, MA, USA). The mix was incubated at 37 °C for 1 h, followed by enzyme inactivation at 95 °C for 5 min. The sequence of the RT-primer was 5′-CAGGTCCAGTTTTTTTTTTTTTTTVN, where V is A, C, and G and N is A, C, G, and T.

### 2.4. Real-Time RT-PCR

cDNA was amplified by RT-qPCR on a LightCycler480. Quantitative PCR was performed in 10 µL of total volume with 3 µL of cDNA, 5 µL of 2X LightCycler 480 SYBR Green I Master (Roche, Mannheim, Germany), 250 nM of each primer (Sigma-Aldrich) (see Table 1), and 1 µL of H_2_O. The primers were designed using miRprimer software [22,23]. Cycling conditions were 95 °C for 1 min, followed by 55 cycles of 95 °C for 1 min and 65 °C for 30 s. Cycle threshold (Ct) values were calculated with the Lightcycler software package and then normalized with U6 values. Experiments were performed in triplicate. Statistical analysis was performed by the 2^−ΔΔC_T_^ method to compare exposed cells with untreated controls. In all cases, a two-sided *p* < 0.05 was considered statistically significant.

### 2.5. Pathway-Enrichment Analysis

The significantly deregulated miRNAs in each of the evaluated conditions were imported to the miRTarBase 8.0 database. Common target genes were selected and imported to the DAVID database to perform a pathway-enrichment analysis. Pathways with a *p*-value < 0.05 and a fold enrichment higher than 10 were selected.

### 2.6. Validation of the Selected Battery of MicroRNAs as Biomarkers of Long-Term Effects of NMs 

A subset of five miRNAs (miR-23a, miR-25, miR-96, miR-210 and miR-502) was found to be sufficient to identify the effects induced by TiO_2_NP and long-term MWCNT exposure. To confirm its suitability, its performance was tested with different NMs and cellular backgrounds. Thus, mouse embryonic fibroblast knockouts for the *Ogg1* gene (MEF Ogg1^−/−^ cells; gift from Dr. Deborah Barnes at Cancer Research UK, UK) previously exposed for 12 weeks to 1 μg/mL of zinc oxide nanoparticles (ZnONP) and to 0.1 μg/mL of cobalt nanoparticles (CoNP) were used [4,5]. In addition, BEAS-2B cells previously exposed for 6 weeks to mesoporous silica nanoparticles (MSiNP) at 10 μg/mL (unpublished results) and cerium dioxide nanoparticles (CeO_2_NP) at 2.5 μg/mL [15] were included. Long-term cell culture conditions and miRNA analysis were conducted, as reported in the previous sections.

## 3. Results

To determine the characteristics of the used NMs, TEM and DLS methodologies were applied. Figure 1A show representative TEM images of TiO_2_NP and MWCNT. While MWCNTs present the fibbers-like shape characteristic of nanotubes, TiO_2_NPs show a more spherical size with irregularities. The mean sizes obtained from TEM images are shown in Figure 1B. The values of hydrodynamic radius and zeta potential are also indicated in Figure 1B. The differences in size observed between TEM and DLS for TiO_2_NP are indicative of a certain degree of aggregation shown by this nanomaterial, as observed in TEM figures and in the polydispersion index value (0.4 ± 0.1). In addition, the Z-potential value for TiO_2_NP (−25.4 ± 0.1 mV) also suggests moderate colloidal stability and the ability to resist aggregation. Similarly, the results obtained for MWCNT also indicate a certain degree of aggregation.

The concentrations selected for TiO_2_NP and MWCNT were based on previous toxicity studies, where they did not exert significant effects on cell viability [18,19]. To confirm that TiO_2_NP and MWCNT were not toxic at the tested concentration in our long-term exposure model, the viability of the exposed cells at week 3 and week 6 was assessed, and the obtained results (Figure 2) indicate that there was no significant decrease in cell viability associated with the treatments or exposure times.

### 3.1. MicroRNA Expression Changes after TiO_2_ NP Exposure

Figure 3 shows the analysis of the observed response after TiO_2_NP exposure at the highest concentration used. The expression analysis at the lowest concentration is shown in Appendix A. Among the entire panel of miRNAs, 29% were significantly deregulated at both tested time points: 41.9% at week 3 and 29% at week 6.

Among those deregulated at both time-points, most miRNAs showed an overexpression compared to the time-matched controls. The six most upregulated miRNAs were miR-23a, miR-25, miR-199b, miR-210, miR-505, and miR-1271. For instance, miR-505 and miR-1271 showed overexpression of 86.21 ± 0.26 and 13.10 ± 0.20 folds at week 3 and 6.48 ± 0.23 and 29.15 ± 0.27 folds at week 6, respectively. On the contrary, only miR-541 appeared downregulated at both weeks [0.2 ± 0.13 week 3; 0.3 ± 0.25 week 6]. Among those miRNAs showing significant changes only at week 6 (Figure 3B), miR-96 was the most upregulated (38.43 ± 0.13 folds). As for week 3, miR-148b, miR-200a, and miR-939 were found to be highly upregulated. In addition, there were some miRNAs showing different behaviors, depending on the evaluated sampling point (week). Thus, while miR-154 and miR-200b were overexpressed at week 3 and downregulated at week 6, miR-486 was inhibited at week 3 but underwent a significant increase in expression at week 6.

### 3.2. MicroRNA Expression Changes after MWCNT Exposure

Those miRNAs deregulated after exposure to the highest concentration of MWCNT are represented in Figure 4A. The expression analysis at the lowest concentration is shown in Appendix A. Figure 4B represents those miRNAs that were significantly deregulated at week 3, at week 6, or at both exposure times.

Most miRNAs were deregulated in both stages of the exposure, with only miR-21, miR-23a, miR-96, and miR-210 being overexpressed. From them, miR-21 and miR-210 reached an overexpression of at least 16.23 ± 0.16 and 45 ± 0.43 folds at week 3 and 18.07 ± 0.53 and 11.87 ± 0.99 folds at week 6, respectively. However, most of them presented under-expression at both times (miR-155, miR-200b, miR-222, miR-342, miR-541, miR-939, and miR-1271). Among those that were inhibited, miR-541 and miR-939 showed an expression of 0.21 ± 0.14 and 0.04 ± 0.17 folds at week 3 and 0.17 ± 0.04 and 0.04 ± 0.15 folds at week 6, respectively. On the other hand, miR-25, miR-31, miR-154, miR-199b, miR-200a, miR-218, miR-486, and miR-505 were only deregulated at week 6 of exposure. Among them, miR-154, miR-200a, and miR-505 were the only ones downregulated. On the contrary, at week 3, most were downregulated (miR-148b, miR-200c, miR-203a, and miR-218), except for miR-200a. This miRNA showed a different expression behavior, depending on the observed exposure time; thus, while at early stages, it was upregulated, it was under-expressed at the late stages. The opposite happened with miR-218, which was downregulated at week 3 and upregulated at week 6.

### 3.3. Pathway-Enrichment Analysis

Figure 5 represents those functional pathways enriched when DAVID analysis was carried out. This analysis was performed based on the potential common target genes from the miRTarBase of those miRNAs significantly deregulated at week 3 and week 6 after the exposure to TiO_2_NP (A) and MWCNT (B). The enriched pathways after TiO_2_NP exposure (week 3) were: (i) cell-cycle arrest and (ii) signal transduction (NIK/NF-kappaB, FGF, and TGFβ signaling pathways). On the other hand, the most significantly enriched pathways at week 6 were (i) cell-death programs (senescence, apoptosis, autophagy), (ii) cell-cycle arrest, (iii) heat-shock response, (iv) inflammatory response, (v) cell migration, and (vi) signal transduction (growth factor and interferon-gamma-signaling pathways). Regarding enriched pathways after MWCNT exposure at week 3, they were (i) cell-cycle arrest, (ii) signal transduction (Wnt and epidermal-growth-factor-signaling pathways), (iii) heat-shock response, and (iv) adherent junctions. Finally, those enriched at week 6 were (i) cell-cycle arrest, (ii) cellular senescence, (iii) angiogenesis, (iv) cell migration, and (v) signal transduction (TGFβ signaling pathway).

### 3.4. Validation of a Set of MicroRNAs as Informative Biomarkers of Long-Term Effects of NMs 

From the analysis of the results obtained in the expression of our initial battery of miRNAs, we propose a small set of miRNAs as potential biomarkers of effect after exposure to NMs. Thus, miR-23a, miR-96, and miR-210 were initially included because they were overexpressed in all concentrations and times of the exposure to both NMs (TiO_2_NP and MWCNT). In addition, miR-25 and miR-502 were added to the list because they were overexpressed in all concentrations and times of at least one NM—in our case, after TiO_2_NP exposure. To validate the usefulness of the five proposed miRNAs, we checked their response in other cells following previous long-term exposure to other NMs, namely CoNP, ZnONP, MSiNP, and CeO_2_NP. The obtained results are indicated in Figure 6. As observed, all five candidate miRNAs showed significant overexpression in all tested conditions, i.e., CoNP exposure the condition inducing the greater response.

Finally, a pathway-enrichment analysis was performed with the common target genes of this selected/proposed set of miRNAs. The three most enriched pathways were (i) signal transduction (PI3k, IGF-1, Wnt, mTOR, and FoxO-signaling pathways), (ii) adherent junctions, and (iii) cell-cycle arrest (Figure 7). Thus, most miRNAs from the selected set of miRNAs act as controlling genes from the three functional categories. However, two of the selected miRNAs (miR-23a and miR-502) were only present in the (i) signal-transduction and (ii) cell-cycle-arrest categories. Additionally, each pathway is controlled by several target genes, and the overlapping area among the three pathways indicates that EP300 controls the three pathways. At the same time, each one of the genes is controlled by several miRNAs, as indicated by the colored bar below the target gene.

## 4. Discussion

At present, enough pieces of evidence demonstrate that exposure to NMs can lead to a wide set of harmful effects, including inflammatory response, DNA damage, oxidative stress, lipid peroxidation, apoptosis, altered gene expression, immunotoxicity, reproductive toxicity, and carcinogenesis [24,25]. Most of the studies generating such information have been conducted using in vitro methods as suitable approaches to in vivo-induced effects. However, most of these studies are far away from simulating real human exposures scenarios since they use short-term exposures and high (and non-biologically relevant) concentrations. Therefore, there is a need to implement in vitro chronic-exposure models in the field and to use them to assay low and non-cytotoxic/concentrations of the NMs under study [26]. An additional advantage of in vitro long-term exposure systems is that they permit the detection of different long-term effects, such as those related to the acquisition of a tumoral phenotype. Indeed, the cell-transformation ability of different NMs has already been reported using these experimental approaches, as demonstrated with cobalt nanoparticles [4], titanium dioxide nanoparticles [19], multiwalled carbon nanotubes [20], nickel oxide nanoparticles [26], and cerium oxide [15].

Epigenetic changes are among the different biomarkers associated with cell transformation processes [11], including miRNA expression changes [27]. In the present study, BEAS-2B cells following long-term exposure to TiO_2_NP and MWCNT showed altered expression changes in a battery of 33 miRNAs related to inflammation, cellular stress, or the carcinogenesis processes. Thus, our system revealed a large impact on miRNA expression in cells exposed to both NMs. TiO_2_NP exposure triggered overexpression of miR-21, miR-148b, miR-154, miR-200a, miR-200b, miR-218, and miR-939 at the early stages of the exposure. These miRNAs target genes implicated in cell-cycle arrest, signal transduction, and adherent junctions. It is worth highlighting that miR-21 targets some common tumor-suppressor genes, such as *PTEN* or *PDCD4* [28]. The *PTEN* gene plays a key role in regulation of the cell cycle, inhibiting cell growth and division at the protein level [29]. Moreover, an elevated expression of miR-21 has been related to poor prognosis in many types of cancers [30]. After 6 weeks of exposure, a large number of miRNAs appeared to be downregulated (miR-30c, miR-30d, miR-132, miR-135b, miR-154, miR-155, miR-200b, and miR-342), while others were overexpressed (miR-31, miR-34a, miR-96, and miR-132). Interestingly, these miRNAs control pathways related to cell-death programs, cell-cycle arrest, heat-shock response, inflammatory response, cell migration, and signal transduction.

Regarding the effects observed under long-term exposures to MWCNT, miR-148b, miR-200c, miR-203a, and miR-218 showed under-expression at week 3. They all mostly control pathways related to cell-cycle arrest, signal transduction, heat-shock response, and adherent junctions. After 6 weeks of exposure, most of the miRNAs were observed to be upregulated, such as miR-25, miR-31, miR-199b, miR-218, miR-486, while miR-154, miR-200a, and miR-505 were found to be downregulated. All of them were related to cell-cycle arrest pathways, cell senescence, angiogenesis, cell migration, and signal-transduction pathways.

One of the conclusions reached from the analysis of the effects of this wide battery of 33 miRNAs is that a few of them can be representative of the epigenetic alterations induced by nanomaterials. Accordingly, five miRNAs, namely miR-23a, miR-25, miR-96, miR-210, and miR-502, were proposed to be used as a set of biomarkers of effect when the health consequences of different nanomaterials are evaluated. Among them, and according to the literature, miR-23a potentiates the epithelial-to-mesenchymal transition process (EMT), downregulating E-cadherin and increasing expression of vimentin, two key proteins in the EMT process [31]. Similarly, another study found that miR-23a was overexpressed in exosomes derived from A549 human lung adenocarcinoma cells, inducing EMT [32]. Moreover, this process has been related to metastasis and poor prognosis in lung cancer patients [33]. Regarding miR-25, its upregulation was correlated with lymph node metastasis, with a poor prognosis in non-small-cell lung cancer (NSCLC) patients [34].

Regarding the role of specific miRNAs in the regulation of the tumoral process, they can act as oncomiRs or as tumor-suppressor miRs. According to the literature, the five miRNAs selected in our study act as oncomiRs [35,36]. The overexpression of miR-96 after all exposures correlates with studies that evidence its role as oncomiR. These studies demonstrate that the overexpression of miR-96 promotes proliferation and invasion and inhibits apoptosis in NSCLC by targeting different genes, such as *LMO7, RECK,* and *FOXO3* [37,38,39]. Additionally, it is known that miRNAs can interchange their roles as oncomiRs or tumor-suppressor miRs, according to the tumor type. They can simultaneously produce promoting and tumor-suppressive effects. The balance between their effects will determine whether a specific miRNA produces a net oncogenic or net tumor-suppressive effect. At present, there is conflicting literature as to whether specific miRNAs are oncogenic or tumor-suppressive [40]. As an example, miR-1271 interchanges its role between either oncomiR or tumor suppressor miR, depending on the tumor type and even depending on the target that we observe within the same type of cancer. Similar behavior has been recently described for miR-1297 [41]. From our results, this miRNA exhibits different behavior, depending on the exposure. While TiO_2_NP exposure triggered its overexpression, MWCNT exposure inhibits its expression.

Interestingly, the goodness of the selected five miRNAs as constituents of a battery useful for evaluation of the cell-transforming potential of NMs was confirmed when applied to transformed cells resulting from previous experiments. These BEAS-2B/MEF cells come from long-term cultures exposed to different nanomaterials (CoNP, ZnONP, MSiNP, and CeO_2_NP) and expressing different biomarkers of cell transformation.

## 5. Conclusions

To conclude, our study evidences that long-term, low-dose exposure to NMs induces miRNA expression changes directly associated with the oncogenic phenotype. Although our data cannot clearly state whether these miRNAs suppose “the spark” needed to start a tumoral process or whether they are symptoms of the tumoral phenotype, we consider that this area is an interesting field requiring further investigation. It should be emphasized that since miRNAs participate in many overlapping cellular processes, the miRNA-level changes of the proposed set can also be conceived of as more general biomarkers for toxicological assessment of NMs under long-term scenarios of exposure. Thus, understanding the link between miRNA expression changes and the long-term effectsinduced by NMs exposure, such as carcinogenesis, can provide valuable information about the underlying mechanism(s), which will benefit NM hazard and risk assessment. To this end, the different pathways identified to be key targets of the proposed set of miRNAs can be considered a promising starting point. In this way, our proposal of using a restricted set of miRNAs can be a powerful tool to determine the potential carcinogenic risk of environmental exposures to NMs. Obviously, the use of this battery can be extended to any agent suspected to have carcinogenic potential.

Exposure to many types of environmental agents has been reported to be able to deregulate the expression of different miRNAs, and such regulation can be used as a biomarker of cancer development induced by environmental factors [42]. Among such environmental factors, plastics and plasticizers are among the most interesting emergent pollutants. In this context, a recent revision indicates that plasticizers alter the expression of different miRNAs. Such a genotoxic/oncogenic response could eventually lead to alterations in the cell signaling pathways involved in different overlapping cellular processes [43]. As a consequence, they can be useful targets in the assessment of the harmful effects of environmental agents, including nanomaterials.

## Figures and Tables

**Figure 1 nanomaterials-11-03458-f001:**
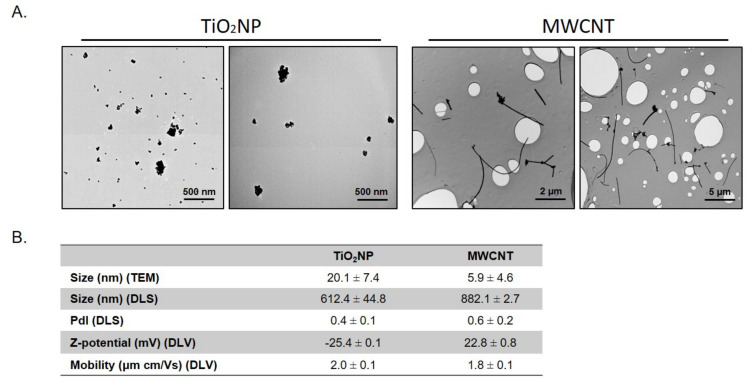
Characterization of TiO_2_NP and MWCNT. (**A**) TEM representative images of both NPs. (**B**) Different parameters, as measures by TEM and Zetasizer. Data are represented as mean ± SD (*n* = 3).

**Figure 2 nanomaterials-11-03458-f002:**
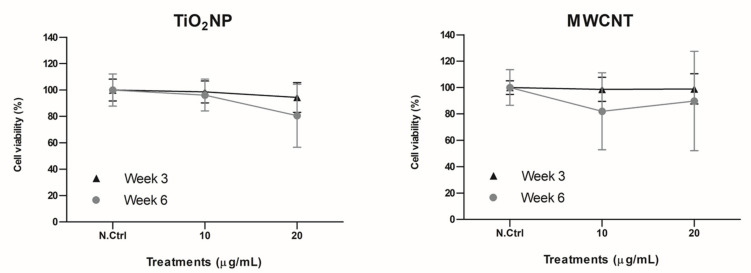
Relative viability of BEAS-2B cells at weeks 3 and 6 of chronic exposure to TiO_2_NP and MWCNT. Viability percentages were calculated by averaging the number of cells counted for each condition in three independent experiments. Cell viability is represented as the percentage of counted cells compared to the untreated time-matched controls ± SEM. Data were analyzed by comparing each condition to the untreated time-matched control (Student’s *t*-test).

**Figure 3 nanomaterials-11-03458-f003:**
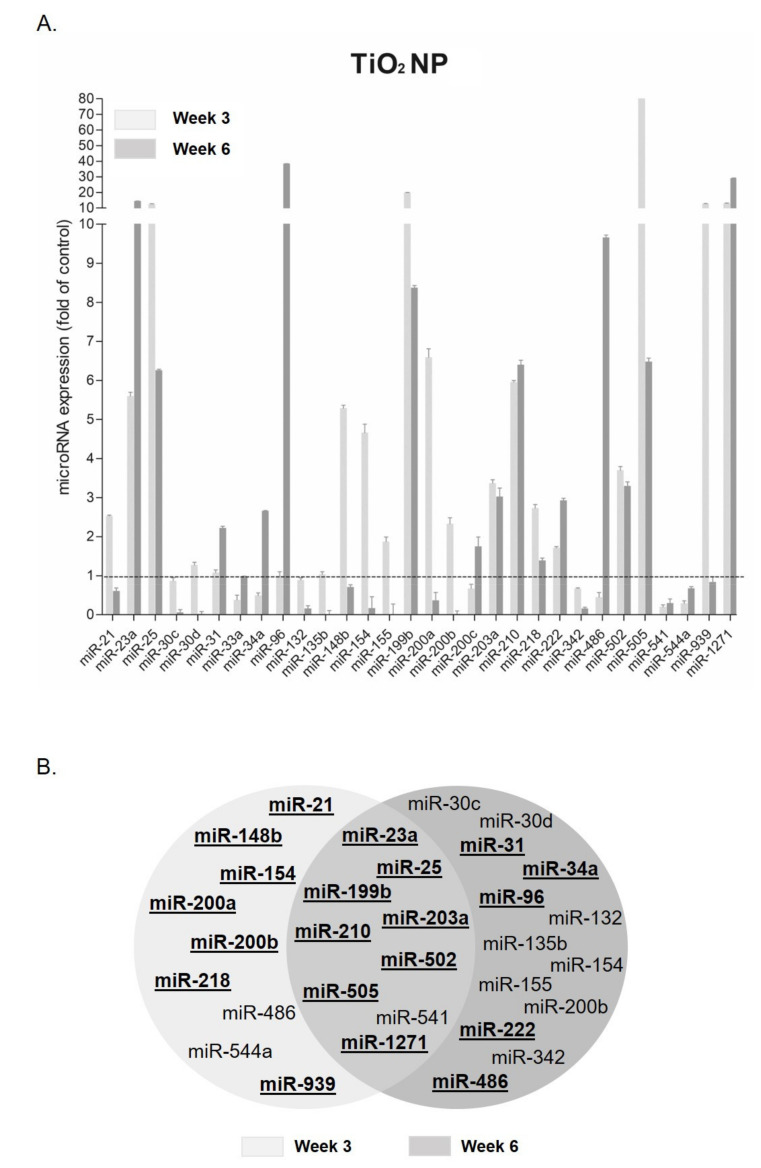
MiRNA expression changes of BEAS-2B cells exposed to the highest concentration (20 μg/mL) of TiO_2_NP. (**A**) Deregulated miRNAs at week 3 and week 6 after exposure to TiO_2_NP. Data are plotted as mean, and error bars represent the SEM. (**B**) Venn diagram showing the number of miRNAs significantly deregulated at week 3 and week 6 of exposure. The overlapping area indicates the number of miRNAs commonly deregulated at both exposure times. Overexpressed miRNAs are in bold and underlined. Results were analyzed with the Student’s *t*-test (*p* < 0.05).

**Figure 4 nanomaterials-11-03458-f004:**
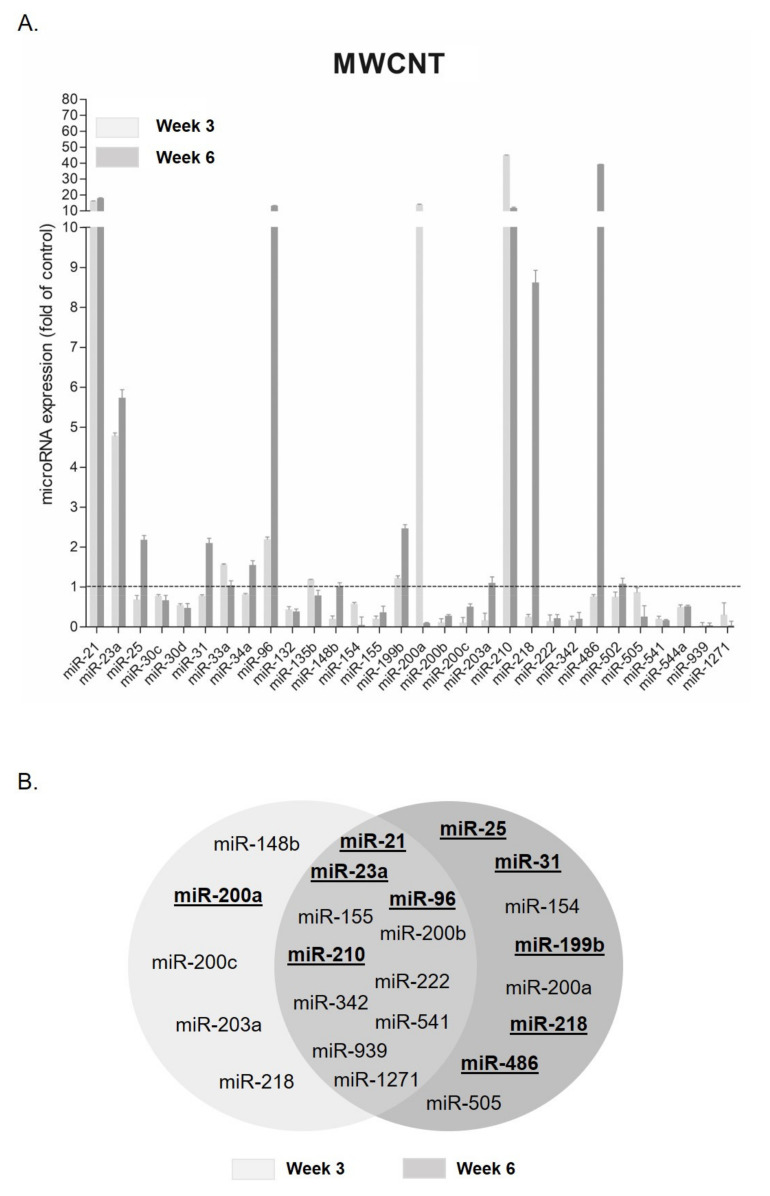
MiRNA expression changes of BEAS-2B cells exposed to the highest concentration (20 μg/mL) of MWCNT. (**A**) Deregulated miRNAs at week 3 and week 6 after the exposure to MWCNT. Data are plotted as mean, and error bars represent the SEM. (**B**) Venn diagram showing the number of miRNAs significantly deregulated at week 3 and week 6 of exposure. The overlapping area indicates the number of miRNAs commonly deregulated at both exposure times. Overexpressed miRNAs are in bold and underlined. Results were analyzed with the Student’s *t*-test (*p* < 0.05).

**Figure 5 nanomaterials-11-03458-f005:**
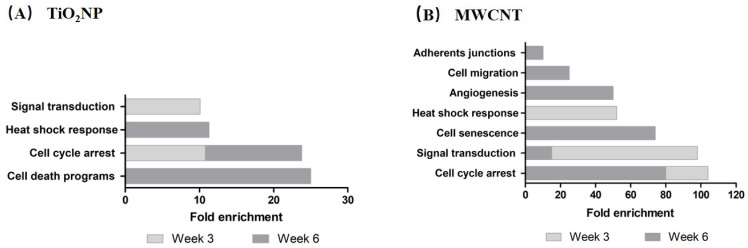
Pathway-enrichment analysis for the target genes of the miRNAs significantly deregulated at week 3 and week 6 after the exposure to (**A**,**B**). Pathways were selected based on *p*-value < 0.05 and fold enrichment higher than 10.

**Figure 6 nanomaterials-11-03458-f006:**
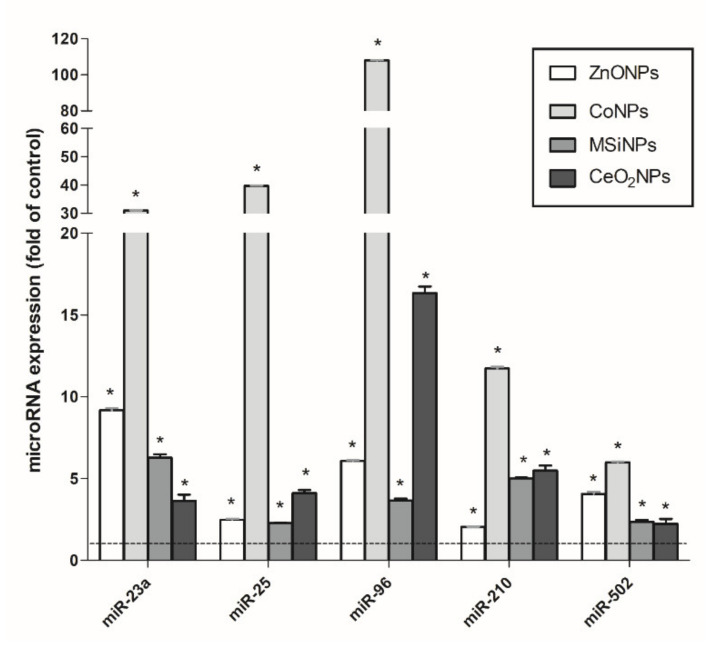
microRNA expression changes in the set of selected miRNAs (miR-23a, miR-25, miR-96, miR-210, miR-502) after the exposure to ZnONP, CoNP, MSiNP, and CeO_2_NP. Data are plotted as mean, and error bars represent the SEM. Results were analyzed with the Student’s *t*-test (* *p* < 0.05).

**Figure 7 nanomaterials-11-03458-f007:**
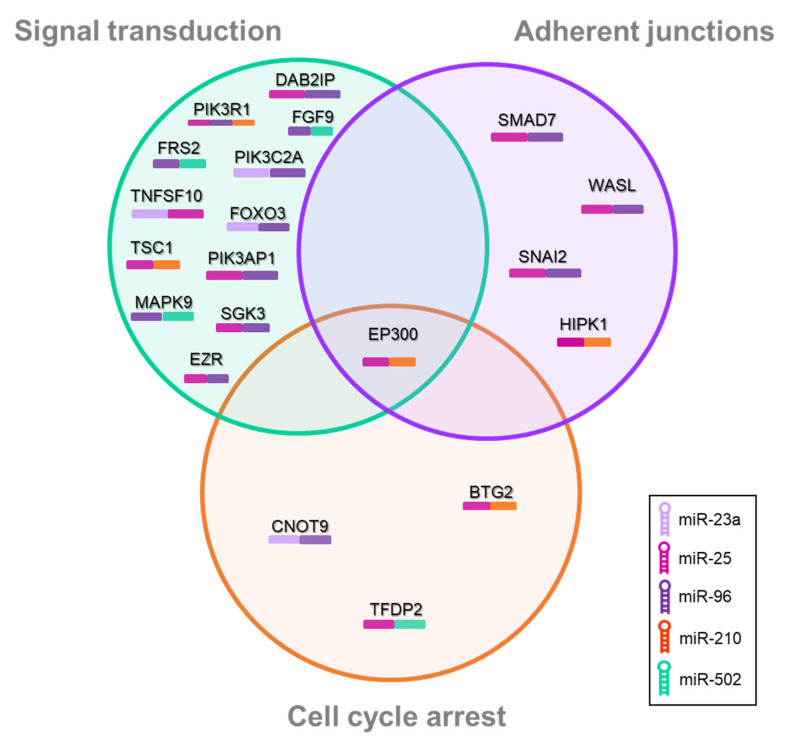
Venn diagram showing the pathway-enrichment analysis for the common target genes of the set of selected miRNAs (miR-23a, miR-25, miR-96, miR-210, miR-502). Signal transduction, adherent junctions, and cell-cycle arrest were the most enriched pathways. Colored bars under target genes represent the miRNAs of the legend. Pathways were selected based on *p*-value < 0.05 and fold enrichment higher than 10.

**Table 1 nanomaterials-11-03458-t001:** MicroRNA primers sequences.

	FORWARD	REVERSE
>hsa-miR-21-5p	TCAGTAGCTTATCAGACTGATG	CGTCCAGTTTTTTTTTTTTTTTCAAC
>hsa-miR-23a-5p	CATCACATTGCCAGGGAT	CGTCCAGTTTTTTTTTTTTTTTGGAA
>hsa-miR-25-3p	CATTGCACTTGTCTCGGT	GGTCCAGTTTTTTTTTTTTTTTCAG
>hsa-miR-30c-5p	GCAGTGTAAACATCCTACACTCT	TCCAGTTTTTTTTTTTTTTTGCTGA
>hsa-miR-30d-5p	AGTGTAAACATCCCCGACT	GGTCCAGTTTTTTTTTTTTTTTCTTC
>hsa-miR-31-5p	GCAGAGGCAAGATGCTG	GTCCAGTTTTTTTTTTTTTTTAGCTATG
>hsa-miR-33a-5p	CGCAGGTGCATTGTAGT	GTCCAGTTTTTTTTTTTTTTTGCAAT
>hsa-miR-34a-5p	GCAGTGGCAGTGTCTTAG	GGTCCAGTTTTTTTTTTTTTTTACAAC
>hsa-miR-96-5p	CAGTTTGGCACTAGCACA	GGTCCAGTTTTTTTTTTTTTTTAGCA
>hsa-miR-124-5p	GCAGCGTGTTCACAGC	TCCAGTTTTTTTTTTTTTTTATCAAGGT
>hsa-miR-126-5p	CGCAGCATTATTACTTTTGGT	CCAGTTTTTTTTTTTTTTTCGCGT
>hsa-miR-132-5p	ACCGTGGCTTTCGATTG	GGTCCAGTTTTTTTTTTTTTTTAGTAAC
>hsa-miR-135b-5p	GCAGTATGGCTTTTCATTCCT	GGTCCAGTTTTTTTTTTTTTTTCACA
>hsa-miR-148b-3p	GCAGTCAGTGCATCACAGA	GGTCCAGTTTTTTTTTTTTTTTACAAAG
>hsa-miR-154-5p	GCAGTAGGTTATCCGTGTTG	GTCCAGTTTTTTTTTTTTTTTCGAAG
>hsa-miR-155-5p	CGCAGTTAATGCTAATCGTGATAG	AGGTCCAGTTTTTTTTTTTTTTTACC
>hsa-miR-199b-5p	CAGCCCAGTGTTTAGACTATC	GTCCAGTTTTTTTTTTTTTTTGAACAG
>hsa-miR-200a	AGCATCTTACCGGACAGT	CCAGTTTTTTTTTTTTTTTCCAGCA
>hsa-miR-200b-5p	GCATCTTACTGGGCAGCA	GGTCCAGTTTTTTTTTTTTTTTCCAA
>hsa-miR-200c	CGTCTTACCCAGCAGTGT	GGTCCAGTTTTTTTTTTTTTTTCCA
>hsa-miR-203a-3p	CAGGTGAAATGTTTAGGACCA	GGTCCAGTTTTTTTTTTTTTTTCTAGT
>hsa-miR-210-5p	TGCCCACCGCACA	GGTCCAGTTTTTTTTTTTTTTTCAGT
>hsa-miR-218-5p	CGCAGTTGTGCTTGATCT	TCCAGTTTTTTTTTTTTTTTACATGGT
>hsa-miR-222-3p	GCAGAGCTACATCTGGCT	CCAGTTTTTTTTTTTTTTTACCCAGT
>hsa-miR-224-5p	GCAGCAAGTCACTAGTGGT	TCCAGTTTTTTTTTTTTTTTAACGGA
>hsa-miR-342-3p	GTCTCACACAGAAATCGCA	GGTCCAGTTTTTTTTTTTTTTTACG
>hsa-miR-486-3p	GGGGCAGCTCAGTACA	GGTCCAGTTTTTTTTTTTTTTTATCCT
>hsa-miR-502-3p	AATGCACCTGGGCAAG	GGTCCAGTTTTTTTTTTTTTTTGAATC
>hsa-miR-505-3p	CGTCAACACTTGCTGGT	GGTCCAGTTTTTTTTTTTTTTTAGGA
>hsa-miR-541-3p	GTGGTGGGCACAGAATC	CCAGTTTTTTTTTTTTTTTAGTCCAG
hsa-miR-544a	GCAGATTCTGCATTTTTAGCAAG	GGTCCAGTTTTTTTTTTTTTTTGAAC
>hsa-miR-939-3p	CCTGGGCCTCTGCTC	GGTCCAGTTTTTTTTTTTTTTTCTG
>hsa-miR-1271-3p	TGCCTGCTATGTGCCA	TCCAGTTTTTTTTTTTTTTTGCCT
>U6 small nuclear RNA	CTCGCTTCGGCAGCACA	AACGCTTCACGAATTTGCGT

## Data Availability

Not applicable.

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
