# Peer review of "MicroRNAs as a Suitable Biomarker to Detect the Effects of Long-Term Exposures to Nanomaterials. Studies on TiO2NP and MWCNT"

_nanomaterials, 2021, doi:10.3390/nano11123458_

Round 1

Reviewer 1 Report

The manuscript by Ballesteros et al. entitled „microRNAs as a suitable biomarker to detect the effects of long-term exposure to nanomaterials. Studies on TiO2NPS and MWCNTs.” describes evaluation of expression of a panel of 33 miRNAs in BEAS-2B cells treated with titanium and carbon nanomaterials. In my opinion, the manuscript has serious drawbacks and is not suitable for publication in Nanomaterials.

Specific comments:

1) First of all, the conclusion that selected five miRNAs are suitable biomarker of long-term exposure to nanoparticles or nanoparticles-induced tumorigenesis is not justified. Chemical carcinogens should be used as a positive control and the results for chemical carcinogens and nanoparticles should be compared to check if observed miRNA expression changes are specific for nanoparticles-induced tumorigenesis or just for tumorigenesis in general. Moreover, the experiment were performed in only two cell lines and it is unknown if they are applicable for other cell types.

2) The experimental setup is strange and should be explained. Cells were initially treated for 4 weeks then frozen, then thawed and treated for another 2 weeks. Why were the cells frozen?

3) How was the viability test performed? It should be described in the Materials and Methods section. Description in Fig. 2 legend is confounding e.g. if the cell viability is represented as a percentage of untreated control then the value for untreated control should be 100% which is not.

4) According to table in fig1B, TiO2NPS seen on TEM have size of about 20 nm. But the image in figure 1A shows that they are much larger (200 nm?).

5) It should be clarified if DLS measurements and TEM imaging were performed in the presence of BSA. Fig 1 legend suggests so, but section 2.1 “Nanomaterials characterization” says nothing about it.

6) The degree of agglomeration of nanoparticles under study is high. The authors should prevent this agglomeration (e.g. by coating of nanoparticles) and repeat the experiments.

7) Page 5, lines 171-172: “The viability of the exposed cells at week 3 (1+2)…” – Why 1+2?

8) Page 12, lines 328-329: “miR-21 targets some common tumor suppressor genes, such as PTEN or PDCD4” – Are PTEN and PDCD4 predicted or confirmed targets of miR-21?

Author Response

see the attached document

Reviewer 2 Report

The main goal of the study was to evaluate the effect of long-term exposure to TiO2 NP and MWCNT upon the levels of 33 microRNAs related to a transformed phenotype in BEAS-2B cells. Based on the obtained data a subset of five microRNAs were selected and their usefulness as biomarkers of transforming effects tested in BEAS-2B cells exposed to CoNP, ZnONP, MSiNP and CeO2NP). All the five microRNAs were found to be upregulated in the exposed cells and the authors conclude that this battery of microRNAs is suitable for predicting the potential transforming ability of NM. While the subject is of great relevance and interest for Nanomaterials audience, the manuscript can be substantially improved, and some aspects must be clarified.

General comments:

It is not clear whether the authors consider the observed microRNAs changes to be or not specific to NM (all types?) exposure.

The authors must made clear the number of independent experiments (and replicate number) of the different assays performed.

Have the authors investigated or detected any interference of the tested NM in the performed assays?

Specific comments:

Title

Please consider replacing the “TiO2NPs” and “MWCNTs” to “TiO2NP” and “MWCNT”, respectively.

Abbreviations should be defined in their first appearance in the text. The authors should revise this aspect throughout the text.

Introduction

Page 2, lines 48-51

In vitro long-term exposures to low doses of nanomaterials have already been shown to induce the expression of different hallmarks of cancer phenotype, as observed in exposures to cobalt nanoparticles [4], zinc oxide nanoparticles [5], and silver nanoparticles [6].

The authors can also include a reference to carbon-based NM as MWCNT were tested in the present study.

Page 2, lines 79-81

“This study has gone one step further than the previous one, proposing a reduced but accurate battery of microRNAs useful to detect potential tumoral effects induced by NMs’ exposures.”

Please add the reference of the cited work.

Materials and Methods

The employed methodologies and techniques can be better described, namely NM characterization and the exposure procedures. I would even suggest including a schematic representation of the experimental protocol.

The authors should replace the term “treatment” by “exposure”.

In ref. #20, the authors must provide the right link to the NanoGenotox dispersion protocol document.

Page 2, lines 92-94

“For the cell treatments, TiO2NPs and MWCNTs were pre-wetted in 0.5% absolute ethanol and dispersed in 0.05% bovine serum albumin (BSA) in MilliQ water.”

Are the authors referring to the stock or working suspensions?

Two important aspects regarding cell culture and exposure must be clarified: the seeding density, when do cells reach confluence and if the mass of NM/area was maintained constant in all the performed assays, an important issue in nanomaterial’s testing. The tested concentrations should be expressed in terms of mass of NM/cm2 and not in mass/mL.

Moreover, it is not clear how have the tested NM been prepared for cell incubation. This aspect must be better explained in the text.

A major weakness of the study is that NM characterization has only been performed in BSA and not in cell incubation medium, whose composition is also missing in the text.

A statistical analysis section can be included.

Results

The results can also be better described and explored.

Page 4, lines 155-157

“These were determined by measuring 100 randomly nanoparticles in several TEM images using ImageJ. Specifically, the mean size of MWCNTs was calculated by measuring the width of isolated fibers.”

This info should be moved to the Materials and Methods section.

How have the authors assessed cell viability? Have the authors used a positive control?

I would suggest replacing Figures 3A, 4A, S1A and S1B by a single Table. This way, would be much easier to the reader to compare the effect of time, concentration and NM type in microRNAs levels. The Veen diagrams can be merged into a single figure.

Have the microRNA levels in control cells varied between week 3 and week 6?

Please replace “Adherent junctions” by “Adherens junctions”. In Figure 5, for each NM, data from 3 and 6 weeks can also be merged into the same graph.

What was the rational for the choice of fold enrichment higher than 10? Nevertheless, some of the pathways showed in Figure 5 exhibit values lower than 10.

Page 10, lines 272-274

“A further selected microRNA is miR-96; although it was not overexpressed in all doses and times, there was always a positive correlation between the expression levels and the exposure, for both nanomaterials.”

Do authors mean “…a positive correlation between the expression levels at week 6 and the exposure, for both nanomaterials.”

In the validation experiments, two cell lines [MEF (mouse embryonic fibroblasts?) and BEAE-2B] and different exposure times were assessed. Couldn’t these changes have an impact in microRNA expression and make comparisons (and data interpretation) with the original protocol difficult? The authors must discuss it.

Author Response

see the attached document

Reviewer 3 Report

The manuscript by Ballesteros et al., entitled “microRNAs as a suitable biomarker to detect the effects of long-term exposures to nanomaterials. Studies on TiO2NPS and MWCNTs” aimed to link the acquisition of a tumoral phenotype and the expression of a set of miRNAs.

The first impression is, many portions of the manuscript are very poorly written and composed. The manuscript contains several grammatical and writing mistakes that need to be corrected. I, therefore, strongly suggest getting the manuscript reviewed by a professional English scientist. Several sentences are difficult to understand and at times make no sense, while some sentences are confusing. At times, the logical flow is also missing. Below are my comments:

Line 42-44: Although many biological endpoints have been used to evaluate the potentially harmful effects of NMs, few studies have been addressed to evaluate their potential carcinogenic risks using in vitro approaches…….This sentence is a bit confusing. There are several studies that reported the carcinogenic effects of different nanoparticles.

Line 51: Apart from the above-indicated mechanisms…which mechanisms did you mean?

Line 55-58: Interestingly, different studies have demonstrated that many cancers have alternative miRNA expression profiles when they are compared to their normal tissues, showing that significant variability in the initiation, the proliferation, and the control of the apoptotic program of tumor cells is due to numerous epigenetic processes, including the level of expression of miRNAs [13]….complicated sentence. Make it simple.

Line 64-65: Thus, the expression levels of about 30% of the used miRNA showed relevant changes, including those miRNAs acting both as oncogenes (OncomiRs) or as tumor suppressors (Anti-OncomiRs)….what do you mean by used miRNA?

Line 66-67: This miRNAs classification depends on if they target tumor suppressor genes or oncogenes, respectively [15]….incorrect sentence, re-write.

Line 68-69: From our previous study, we proposed that such a panel of miRNAs should be used as potential biomarkers when the carcinogenic potential of nanomaterials is evaluated…briefly explain here, why?

Line 69-70: To reinforce our proposal here we have evaluated the expression levels of our proposed miRNAs panel in BEAS-2B cells long-term (4 weeks) exposed to both TiO2NPs and MWCNTs…mention why did you select this set of miRNAs, why these are relevant to the study?

72-73: Under such exposure conditions, a tumoral phenotype was induced, according to the soft agar assay, which detects anchorage-independent cell growth…under what exposure/condition? Rephrase the sentence.

Line 73-74: Thus, we aimed to link the adquisition of a tumoral phenotype and the expression of a set of miRNAs…incorrect sentence

Line 86-87: Titanium dioxide nanoparticles (TiO2NPs, NM102) and multi-walled carbonano tubes (MWCNT, NM401)…it is carbon nanotubes, correct it.

Line 90-91: The dynamic light scattering (DLS), and laser Doppler velocimetry (LDV) methodologies (Malvern Zetasizer Nano-ZS zen3600 instrument)….incomplete sentence

Line 93: For the cell treatments, TiO2NPs and MWCNTs were pre-wetted in 0.5% absolute ethanol…what is 0.5% absolute ethanol?

Line 102-104: During the chronic exposure, the culture medium was changed every two days with a new medium containing the same concentrations of TiO2NPs or MWCNT, after two washes with 1% PBS. Weekly cells were sub-cultured by seeding 500,000 cells…check for writing errors/rephrase

Line 105-107: Once the cells were defrosted, they were exposed for two weeks more, following the previous treatment description. In parallel, untreated controls were carried out, and every condition was performed in duplicate…re-write the sentences

Do the authors have any images of the cells before/after treatments with NMs?

Did you check if the NMs were taken up by the cells? Or excreted the effects from outside the cells via surface ligands?

Were the cells healthy after the long treatments? What was the apoptosis/necrosis status? Did you check?

If the exposure to NMs is withdrawn, is it expected that the miRNA levels will get back to their original level? What do the authors think? I understand that the authors might not have checked this, I just wanted to learn about the consequences from a human health perspective, if the authors can throw some light on this.

Line 142: To confirm the suitability of the proposed battery we evaluated its response in front of new NMs…re-write the sentence

Line 143-144: Thus, MEF cells previously exposed for 12 weeks to zinc oxide nanoparticles (ZnONPs) at 1 μg/mL, and cobalt nanoparticles (CoNPs) at 0.1 μg/mL were used [4,5]…where the MEF cells come from and why?

Did you use any negative and positive control for the miRNA expression analysis experiment?

Are you aware of any study that reports the changes of these miRNA levels during any other health conditions? My question is how could you co-relate the changes in the level of these selected miRNAs with only exposure to NMs, particularly in humans? In cells, exposed to NMs only, it’s understandable.

Author Response

see the attached document

Round 2

Reviewer 1 Report

All the points raised in my first review have been addressed by the authors. I'm satisfied with the corrections. The manuscript is written much more clearly than before and can be accepted for publication.

Author Response

see the attached document

Reviewer 2 Report

Comments to the authors:

It is not clear whether authors consider the observed microRNAs changes to be or not specific to NM (all types?) exposure.

Authors RESPONSE:

Our proposal is that the proposed battery can be used after exposure to any type of NM. This approach would avoid using a wide battery, as we initially used.

This point has been also raised by Reviewer#1. Since no chemical carcinogens were used as positive controls and the results for chemical carcinogens and nanoparticles compared to check if the observed miRNA expression changes would be specific for nanoparticles-induced tumorigenesis or just for tumorigenesis in general, the authors cannot categorically claim that the proposed battery is specific for NM, irrespective of its type. Accordingly, the authors must point out this fact in the discussion.

The authors must made clear the number of independent experiments (and replicate number) of the different assays performed.

Authors RESPONSE:

In all cases, three independent experiments with three technical replicates were included. This information is now better explained in section 2.2.

Section 2.2 (2.2. Cell culture and exposure conditions), page 3, line 121: “All conditions were performed in triplicates.” Please clearly indicate in the text the number of independent experiments and technical replicates used.

Materials and Methods

The employed methodologies and techniques can be better described, namely NM characterization and the exposure procedures. I would even suggest including a schematic representation of the experimental protocol.

RESPONSE:

The part corresponding to the NM characterization has been modified for a better understanding (section 2.1). Similarly, the section 2.2 regarding cell exposure has been significantly improved.

Page 3, lines 107-108: “For TEM and Zetasizer measurements the stock solution was dispersed in water and in culture medium, respectively.” Please replace the term “solution” by “suspension”.

Page 3, lines 114-115: “…sub-confluent cells were passaged weekly at a cell density of 5x105…” Per cm2? Per T75 flask? Please clarify.

Page 3, lines 119-120: “In all cases, the mass per volume or per area remained constant.” This statement is rather confusing. Said like this, it sounds that the mass of NM/cell was not kept constant throughout the study. Please clarify it.

Results

Page 5, lines 177-178: “In addition, the Z-potential value for TiO2NP (-25.4 ± 0.1 mV), also suggests moderate colloidal stability and the ability to resist aggregation.” This statement is not concordant with TiNO2 size as determined by DLS. Moreover, TEM images to be provided, in particular for TiO2 NP, should be at a magnification and scale bar that allow readers to have a quick and clear view about the size of the NM.

Please replace “Adherent junctions” by “Adherens junctions”. In Figure 5, for each NM, data from 3 and 6 weeks can also be merged into the same graph.

RESPONSE:

Figure 5 has been modified and weeks 3 and 6 placed in the same graph. Also, the term has been replaced.

Please also replace “Adherent junctions” by “Adherens junctions” in Figure 7 image and caption, and throughout the manuscript.

Page 12, lines 323-324: “It is worth highlighting that miR-21 targets some common tumor 323 suppressor genes, such as PTEN or PDCD4.” Please provide a reference(s).

Author Response

see the attached document

Reviewer 3 Report

After the revision, the overall acceptability of the current manuscript has been improved to some extent. The authors have changed/updated the information suggested. Some sentences that required proper clarifications have now been addressed. Some critical information related to miRNAs/methodologies has also been updated, though the authors were unable to provide all the data as recommended that could further improve the scientific clarity. 

Author Response

RESPONSE:

As the reviewer indicates we have changed/updated all those suggested points. In this way, we consider that the manuscript has significantly improved. Perhaps we have not been able to fully update some other aspects, mainly in the Discussion section, that would improve the scientific clarity. We have gone through the Discussion section, again, without finding points requiring further changes to improve their understandability.